



# An annual 30 m cultivated pasture dataset of the Tibetan Plateau from 1988 to 2021

Binghong Han[1], Jian Bi[2], Shengli Tao[3], Tong Yang[2], Yongli Tang[2], Mengshuai Ge[3], Hao Wang[4], Zhenong Jin[3], Jinwei Dong[5], Zhibiao Nan[1], Jin-Sheng He[1,3]

[1] State Key Laboratory of Herbage Improvement and Grassland Agro-Ecosystems, and College of Pastoral Agriculture Science and Technology, Lanzhou University, Lanzhou 730000, China

[2] College of Earth and Environmental Sciences, and Center for Remote Sensing of Ecological Environments in Cold and Arid Regions, Lanzhou University, Lanzhou 730000, China

[3] Institute of Ecology, College of Urban and Environmental Sciences, and Key Laboratory for Earth Surface Processes of the Ministry of Education, Peking University, Beijing 100871, China

[4] State Key Laboratory of Seed Innovation and Grassland Agro-ecosystems, and College of Ecology, Lanzhou University, Lanzhou 730000, China

[5] Institute of Geographic Sciences and Natural Resources Research, Chinese Academy of Sciences, Beijing 100101, China

*Correspondence to*: Jin-Sheng He (jshe@pku.edu.cn) or Jian Bi (bijian@lzu.edu.cn)

**Abstract.** Cultivated pastures have rapidly developed across the Tibetan Plateau over the past several decades, raising concerns about grassland degradation. Accordingly, considerable attention is focused on the protection of Tibetan grassland ecosystems. However, high-resolution spatial distribution of cultivated pastures on the Tibetan Plateau remains poorly understood, primarily due to the difficulty of discriminating cultivated pastures from other land cover types using remote sensing techniques. The absence of such information hinders efficient agricultural and livestock husbandry management, making it challenging to support ecological protection and restoration efforts. Here, we mapped the cultivated pastures on the Tibetan Plateau at a 30-m resolution for the years 1988 to 2021 using Landsat data on the Google Earth Engine (GEE) cloud computing platform. We built a Random Forest (RF) binary classification model with inputs of the spectral-temporal metrics of Landsat data acquired in the growing season, as well as ancillary topographic data. The model was trained using carefully selected training samples and validated against 2,000 independent random reference points in two pilot study regions with different climates and landscapes. The model achieved an overall accuracy of $97.05\% \pm 0.4\%$ and an F1 spatial consistency score of $82.51\% \pm 14.22\%$ (Precision: $90.04\% \pm 6.18\%$, Recall: $76.74\% \pm 9.91\%$), suggesting high confidence in mapping the distribution of cultivated pastures on the plateau. Using the RF model, we then produced a dataset of cultivated pasture maps for the years from 1988 to 2021 for Qinghai Province and the Tibet Autonomous Region on the Tibetan Plateau, covering 77% of the plateau. At both the province and county levels, the cultivated pasture areas estimated in this study matched well with government statistics in recent years. The area of cultivated pastures on the Tibetan Plateau experienced a significant expansion from 0.46 Mha in 1988 to 1.57 Mha in 2021, with the average annual growth of $33.5 \pm 2.5$ Kha. To our knowledge, we are the first to map cultivated pastures on the Tibetan Plateau, and our RF binary classification approach holds promise in identifying cultivated pastures in



other regions of the world, which could prove invaluable for scientists, policymakers, ecological conservation practitioners, and herdsmen. The dataset is available on Zenodo at https://doi.org/10.5281/zenodo.14271782 (Han et al., 2024).

## 1 Introduction

Grasslands on the Tibetan Plateau play essential roles in carbon, water, and nutrient cycles (Chen et al., 2022; Piao et al., 2020; Wang et al., 2022; Zhang et al., 2022), in maintaining biodiversity (He et al., 2024), in mediating energy balances (Chang et al., 2021), and in supporting the livelihoods of millions of pastoralists (Fuglie et al., 2021; Hou et al., 2021). However, under the joint influence of climate change (He et al. 2020; Yao et al., 2022; Zhang et al., 2020) and human activity (Ding et al., 2022; Li et al., 2021; Zhang et al., 2022), Tibetan grasslands face serious degradation problems (Bardgett et al., 2021; Wang et al., 2022; Zhu et al., 2023). Numerous ecological restoration measures have been implemented in the past two decades to address the problem of grassland degradation on the plateau (Bardgett et al., 2021; Li et al., 2020; Zhu et al., 2023) and to improve the welfare of Tibetan pastoral communities (Fuglie et al., 2021; Hou et al., 2021). The establishment of cultivated pastures, which is common in western developed countries (Vroey et al., 2022), is encouraged in developing countries as one of these efforts (Wang and Zhang, 2023).

Cultivated pastures are also known as tame grasslands/pastures (Fisher et al., 2018; McInnes et al., 2015), agricultural grasslands (Zalite et al., 2016), green fodder lands (Yang et al., 2021), or planted pastures (Parente et al., 2017). Cultivated pastures primarily cultivate alfalfa, silage corn, forage oat, ryegrass, or similar crops. The vegetation spectral signals of these cultivated pastures are similar to those of regular croplands during peak growing seasons. However, cultivated pastures are generally harvested before reaching full maturity to optimize nutrient retention and maintain palatability. As a result, the duration of vegetation growth in cultivated pastures is shorter compared to croplands. This discrepancy may lead to noticeable differences in vegetation spectral signals between regular croplands and cultivated pastures at the end of the growing season (Ashourloo et al., 2018; Yang et al., 2021).

Mapping cultivated pastures on the Tibetan Plateau is important for the following reasons. Firstly, cultivated pastures provide substantial amounts of forage for livestock, the main economic income of Tibetan pastoralists (Fuglie et al., 2021; Hou et al., 2021). Secondly, cultivated pastures are essential for the ecological conservation and restoration efforts in this ecologically fragile area by reducing grazing pressure on natural grasslands (Kumar et al., 2019; Fang et al., 2016). Thirdly, encouraging cultivated pastures on the Tibetan Plateau has led to considerable changes in land use and land cover. These cultivated pastures, if well planned, can have significant impacts on ecosystem services and biodiversity conservation (Chen et al., 2021; Dong et al., 2022); if not well planned, they will result in ecosystem degradation that will be difficult to restore in the extreme environments.

Satellite remote sensing is an essential tool for mapping cultivated pastures (McInnes et al., 2015; Ashourloo et al., 2018; Fisher et al., 2018; Yang et al., 2021; Wang et al., 2022). For example, McInnes et al. (2015) used MODIS data to discriminate native and non-native grasslands in a dry mixed prairie in Canada, with an overall accuracy of 73% assessed by independent





validation. Ashourloo et al. (2018) identified alfalfa fields in Iran and the United States using Landsat time series data, and the overall accuracy reached above 90% by cross-validation, although their method did not require a very dense number of valid observations. LiDAR data has been used to distinguish cultivated grasslands from natural grasslands, as in one study in southwestern Saskatchewan, Canada, and an overall accuracy of 96 % was achieved (Fisher et al. 2018). Satellite remote sensing has also been used to map pastoral lands in China. Yang et al. (2021) used Landsat data during the growing season to

map green fodder fields in the northeastern Tibetan Plateau in 2010, 2015, and 2019, and achieved overall accuracies of 94.2%, 93.1%, and 96.6%. They found that the green fodder lands in northeastern Tibetan Plateau expanded from 16.3 km² in 2010 to 136.1 km² in 2019, 7.35 times the initial area. Wang et al. (2022) identified oat pastures in Shandan County of Gansu Province using Sentinel-2 data from 2019 to 2021, with an overall accuracy of 98% assessed by cross-validation. They found that the area of cultivated oat pastures decreased from 347.8 km² in 2019 to 318.9 km² in 2021.

While a number of studies have mapped cultivated pastures (Ashourloo et al., 2018; Fisher et al., 2018; McInnes et al., 2015; Wang et al., 2022; Yang et al., 2021), many mapped cultivated pastures that grow certain types of tame grass species, such as alfalfa (Ashourloo et al., 2018), oat (Wang et al., 2022), and rapeseed (Yang et al., 2021); few studies focused on the mapping of general cultivated pastures, especially in the harsh environments on the Tibetan Plateau. The temporal evolution of the distribution of cultivated pastures on the Tibetan Plateau, which is of great interest to policymakers and researchers,

remains poorly understood.

Therefore, the aims of the study are (1) to develop a method for mapping general cultivated pastures using satellite remote sensing data, (2) to clarify important technical details for successful mapping of general cultivated pastures on the Tibetan Plateau, and (3) to understand the temporal evolution of the spatial distribution of cultivated pastures on the Tibetan Plateau.

## 2 Study region

The Tibetan Plateau spans from 25°59' E to 40°04' E in latitude and from 73°29' N to 104°40' N in longitude, with an average elevation of over 4,000 m (Fig. 1). The region has a continental plateau climate, with an annual mean temperature of 2.0˚C and an annual mean precipitation of 373.5 mm (Zhang et al., 2023). The growing season lasts from April to October (Wang et al., 2020). The land cover types include grasslands, deserts, croplands, forests, etc. (Fig. 1). The Tibetan Plateau is the habitat of over 50 million Tibetan sheep and 13 million yaks (Cheng et al., 2016), which rely on natural grasslands and cultivated

pastures for their forage. The dominant native grass species in this region mainly include *Stipa aliena*, *Carex przewalskii*, and *Kobresia deasyi* (Jia et al., 2019). The cultivated grass species are *Elymus nutans, Medicago sativa, Poa annua, Lolium perenne, Avena sativa* and silage *Zea mays* (Fig. 2).

During the summer of 2021, we conducted a field campaign in Qinghai Province and the Tibet Autonomous Region on the Tibetan Plateau. We travelled 4,280 km and visited the counties of Gonghe, Xinghai, Tongde, Guinan, Guide, Kangma,

Linzhou, Sajia, Dangxiong, Longzi, and Nanmulin. These counties are referred to as the pilot study regions in Qinghai Province and the Tibet Autonomous Region (Fig. 1).

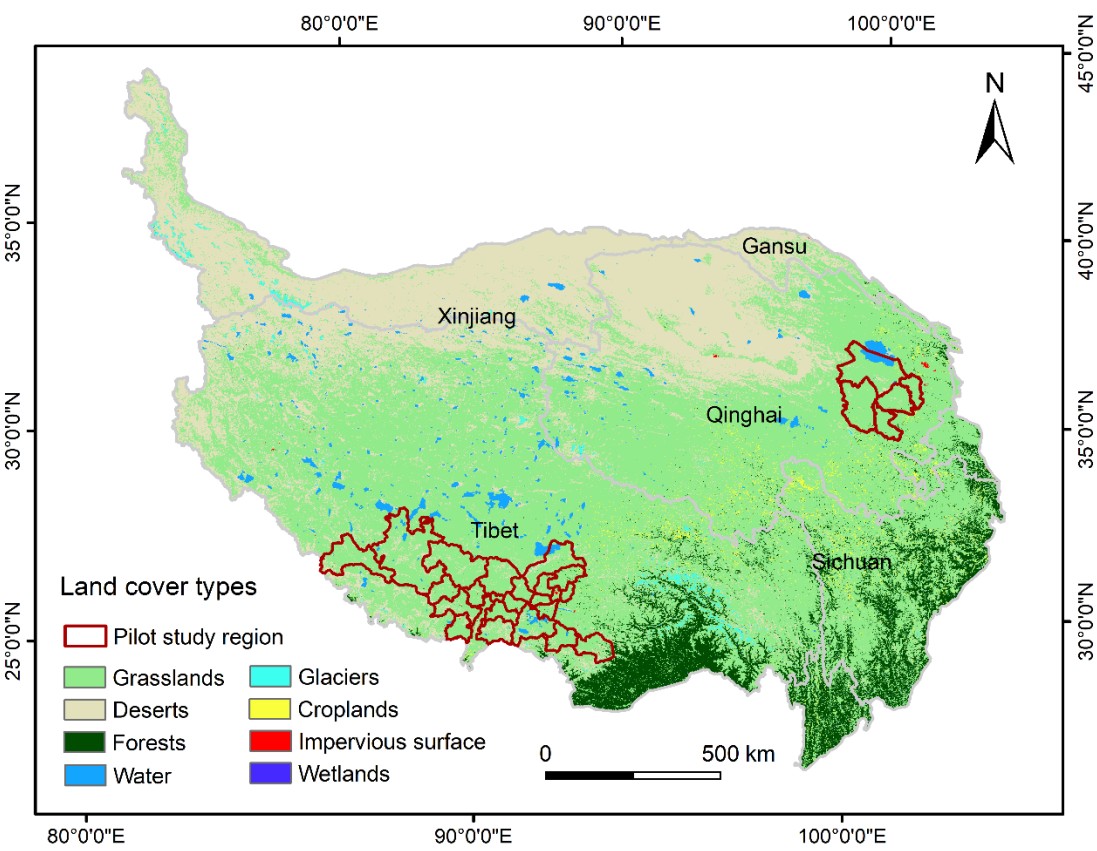

**Figure 1.** The land cover types of the study region and the distribution of the pilot study regions in Qinghai and Tibet. The land cover data source: the Resource and Environment Science and Data Center (*http://www.resdc.cn/*) of the Chinese Academy of Sciences. The binary classification model for mapping cultivated pastures was trained and validated in the pilot study regions.

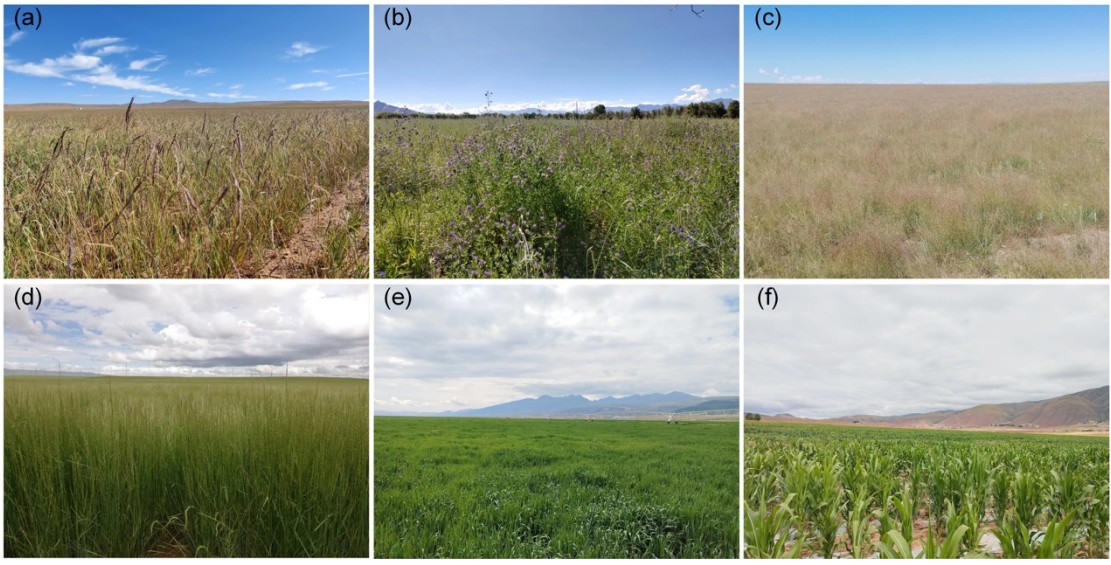



**Figure 2.** Photographs of the cultivated pastures visited in the 2021 field campaign on the Tibetan Plateau. (a) Dahurian wildrye (*Elymus nutans*), (b) alfalfa (*Medicago sativa*), (c) annual bluegrass (*Poa annua*), (d) ryegrass (*Lolium perenne*), (e) oat (*Avena sativa*), (f) silage corn (*Zea mays*).

## 3 Methods

We used surface reflectance (SR) data from the Thematic Mapper (TM), Enhanced Thematic Mapper Plus (ETM+), and Operational Land Imager (OLI) sensors onboard the Landsat satellites (Wulder et al., 2022) to map cultivated pastures on the Tibetan Plateau. The mapping algorithm classified the spectral-temporal metrics of each pixel into two categories: *cultivated pasture* and *other*. *Other* include natural grasslands, forests, croplands, deserts, water bodies, etc. The Random Forest (Breiman, 2001) method was used as the classification algorithm. Since we were interested in the identification of cultivated pastures rather than other land cover types, the RF model was designed as a binary classification model. The Random Forest binary classification model used in this study was trained with representative land cover type samples carefully selected during the field campaign or with the aid of high spatial resolution images on Google Earth. The inputs to the binary classification model were the spectral-temporal metrics of Landsat visible and infrared bands, spectral indices, as well as ancillary topographical information. The binary classification algorithm was implemented on Google Earth Engine (GEE; Gorelick et al., 2017) cloud computing platform using the geoscience data stored on it.

### 3.1 Data

To map cultivated pastures on the Tibetan Plateau, we used the surface reflectance (SR) data from the Landsat 5, 7, and 8 satellites (Roy et al., 2014) in the visible and infrared bands in the years from 1988 to 2021. Globally, the Landsat 5 TM data were available from March 1984 to June 2013, the Landsat 7 ETM+ data available since April 1999, and the Landsat 8 OLI since February 2013. The band settings of OLI are different from those of TM and ETM+, so we used a conversion procedure (Roy et al., 2016) to convert the SR of TM and ETM+ to that of OLI. This way, the RF binary classification model trained with the Landsat 8 data in 2021 as inputs can be applied to historical periods when TM or ETM+ data were available.

The Landsat data have a spatial resolution of 30 meters and a temporal frequency of 16 days (Wulder et al., 2019). The Land Surface Reflectance Code (LaSRC; Vermote et al., 2018) was used to perform atmospheric correction. The data also include Quality Assessment (QA) fields produced with the CFMask method (Zhu and Woodcock, 2014), which label clouds, cloud shadows, snow, water, and pixel saturation. We used Landsat data during the growing season (April to October) on the Tibetan Plateau, as there is little vegetation signal during the non-growing season (Wang et al., 2020). The QA fields in the Landsat data were used to mask out clouds, cloud shadows, snow, and pixel saturation. Although the nominal temporal frequency of Landsat data is 16 days, the actual valid observations tend to have a temporal frequency of more than 16 days due to cloud, cloud shadow, and snow interference. Moreover, due to the side-overlapping of Landsat scenes, some locations have more valid observations compared to others. There were more valid Landsat observations in recent years than in earlier years (Fig. S1).



135 In addition to the SR, we also used seven spectral indices including Normalized Difference Vegetation Index (NDVI; Tucker, 1979), Enhanced Vegetation Index (EVI; Liu and Huete, 1995), Normalized Burn Ratio (NBR; López García and Caselles, 1991), Normalized Difference Built-Up Index (NDBI; Zha et al., 2003), Normalized Difference Phenology Index (NDPI; Wang et al., 2017), Normalized Difference Water Index (NDWI; Gao, 1996), and Modified Normalized Difference Water Index (MNDWI; Xu, 2006). Using these spectral indices can expedite the land cover classification efficiency, at both
140 the training and classification stages.

 Topography can affect the growth conditions of grasses. Cultivated pastures are typically located on flat terrains to facilitate the use of automated machinery for ploughing and harvesting, hence topography features may be useful to identify cultivated pastures. In our study region, we characterized the topography using the Shuttle Radar Topography Mission (SRTM) Digital Elevation Model (DEM) data (Farr et al., 2007), which had a spatial resolution of 30 meters. The RF binary classification
145 model's inputs included slope, aspect, and hill shade derived from elevation.

### 3.2 Spectral-temporal metrics

Remotely sensed vegetation spectrums are characterized by high reflectance in near-infrared wavelengths and low reflectance in visible wavelengths (Tian et al., 2023). During the peak growing season, many vegetation types exhibit similar spectral features (Zeng et al., 2022). We used satellite remote sensing data throughout the growing season because vegetation has
150 different phenological profiles due to species composition and human management (Dong et al., 2016; Parente et al., 2017). We used some key metrics from among all spectral-temporal data to reduce the computation burden while maintaining accuracy (Parente et al., 2019, 2017; Parente and Ferreira, 2018; Wang et al., 2022; Yang et al., 2021). Specifically, we used three descriptive statistical metrics, namely the 25%, 50%, and 75% quantiles of remote sensing data during the growing season instead of the complete time series of all available Landsat data during the growing season (Aghighi et al., 2018; Moon et al.,
155 2021). We referred to these descriptive statistical metrics of remote sensing data as spectral-temporal metrics. The spectral-temporal metrics inherently contained vegetation phenological information that can be used to classify land covers. In summary, we used the 25%, 50%, and 75% quantiles of the time series of all available Landsat visible and infrared SR, seven spectral indices (i.e., NDVI, EVI, NBR, NDBI, NDPI, NDWI, and MNDWI) in the growing season.

### 3.3 The training samples

160 In the pilot study region of Qinghai Province, we visited 81 cultivated pastures. Of these, 40 were Dahurian wildrye (*Elymus nutans*) fields, 11 were miscanthus (*Elymus sibiricus*) fields, 6 were annual bluegrass (*Poa annua*) fields, 4 were ryegrass (*Lolium perenne*) fields, 2 were Guinea grass (*Roegneria grandiglumis*) fields, and 18 were oat (*Avena sativa*) fields. In the pilot study region of the Tibet Autonomous Region, we visited 114 cultivated pastures. Of these fields, 62 were oat pasture (*Avena sativa*), 33 were alfalfa (*Medicago sativa*), 10 were silage corn (*Zea mays*), and 9 were Dahurian wildrye (*Elymus*
165 *nutans*). The detailed number and area of the visited cultivated pastures are listed in Table 1 and shown in Fig. 3a and Fig. 3c. The boundaries of the visited cultivated pastures were recorded using a handheld Global Positioning System (GPS) device.



Fig. 3a and 3c show the spatial distribution of the two land cover categories (*cultivated pasture* and *other*) in the pilot study regions used to train the RF binary classification model. The number and area of the training polygons for *cultivated pasture* and *other* are summarized in Table 2.

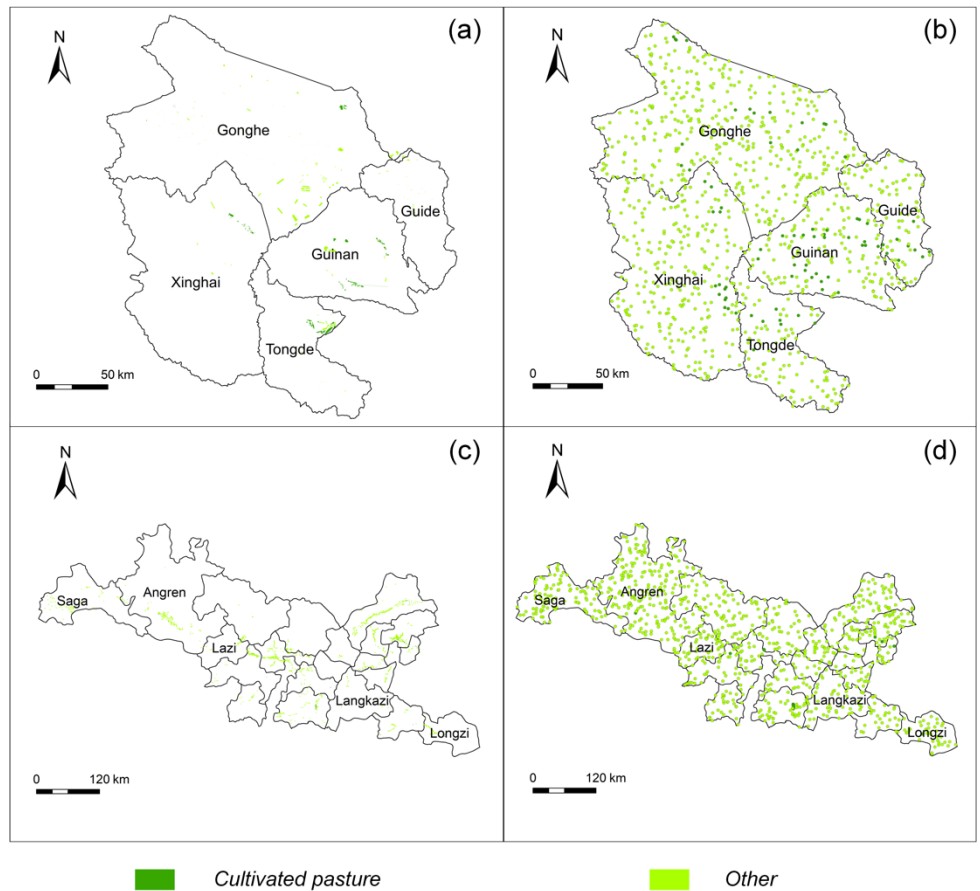

**Figure 3.** The spatial distribution of the training polygons and the validation points in the pilot study regions. The training polygons (a, c) were recorded during the 2021 field campaign, and the 1,000 independent random validation points (b, d) in each pilot study region were labelled with the aid of high-resolution images on Google Earth.

**Table 1.** Summary of the type, number, and area of the cultivated pastures visited during the 2021 summer field campaign in the pilot study

regions.

| Cultivated pasture type | Number of polygons | Number of pixels | Area (ha) |
|---|---|---|---|
| **Qinghai Province** | | | |
| Dahurian wildrye (*Elymus nutans*) | 41 | 40016 | 3601.4 |
| Miscanthus (*Elymus sibiricus*) | 8 | 6931 | 623.8 |
| Annual bluegrass (*Poa annua*) | 8 | 3748 | 337.3 |
| Ryegrass (*Lolium perenne*) | 4 | 3027 | 272.4 |
| Guinea grass (*Roegneria grandiglumis*) | 2 | 1083 | 97.5 |
| Oat (*Avena sativa*) | 18 | 28388 | 2554.9 |



| | | | |
|---|---|---|---|
| Total | 81 | 83193 | 7487.3 |
| **The Tibet Autonomous Region** | | | |
| Oat (*Avena sativa*) | 62 | 13666 | 1229.9 |
| Alfalfa (*Medicago sativa*) | 33 | 2223 | 200.1 |
| Silage corn (*Zea mays*) | 10 | 1539 | 138.5 |
| Dahurian wildrye (*Elymus nutans*) | 9 | 482 | 43.4 |
| Total | 114 | 17910 | 1611.9 |

**Table 2.** Summary of the training polygons in the pilot study regions.

| Region | Land cover category | Number of polygons | Number of pixels | Area (ha) |
|---|---|---|---|---|
| Qinghai Province | *Cultivated pasture* | 81 | 83193 | 7487.4 |
| | *Other* | 465 | 144818 | 13033.6 |
| | Total | 546 | 228011 | 20521.0 |
| The Tibet Autonomous Region | *Cultivated pasture* | 114 | 17910 | 1611.9 |
| | *Other* | 1429 | 31537 | 2838.3 |
| | Total | 1543 | 49447 | 4450.2 |

## 3.4 The government statistics data

China Agricultural Press in Beijing published the annual *China Pratacultural Statistics* (e.g., Li and Wang, 2017), in which
the areas of cultivated pastures were reported at the province level. The area of cultivated pastures at the province level were
summed from the county-level areas of cultivated pastures, which were from the county-level governments' cultivated pasture
census data. However, the areas of cultivated pastures at the county level were not reported in the annual *China Pratacultural
Statistics*.

From 2010 to 2017, the annual *China Pratacultural Statistics* considered all grasslands intervened by humans as cultivated
pastures, including purely cultivated pastures and grasslands improved by human activities (i.e. seed-sowing and grazing
prohibition). From 2001 to 2009 and from 2018 to 2021, the annual *China Pratacultural Statistics* only considered grasslands
with ploughing and seed-sowing management practices as cultivated pastures. The statical caliber in the years from 2010 to
2017 was more reasonable and agreed with our definition of cultivated pastures.

Furthermore, we collected some county-level statistics data (13 counties in Qinghai and 12 counties in Tibet) of the areas
of cultivated pastures for 2021 from the Qinghai Province Bureau of Forestry and Grassland as well as the Bureau of
Agriculture and Rural Affairs of the Tibet Autonomous Region. We collected these county-level statistics data for comparison
with the areas of cultivated pastures mapped through remote sensing.



### 3.5 The binary classification algorithm

A RF binary classification model was used to identify cultivated pastures in the study region (Fig. 4). The classification model consisted of 500 trees and used the spectral-temporal metrics of the remote sensing data (SR in the seven Landsat visible and infrared bands, NDVI, EVI, NBR, NDBI, NDPI, NDWI, and MNDWI) in the growing season as well as the ancillary topographic data as inputs. The RF binary classification model was trained using the training polygons shown in Fig. 3a and Fig. 3c. The training process on the GEE platform took 10 minutes. The RF binary classification model generated the likelihood

of each pixel belonging to the two land cover categories (*cultivated pasture* and *other*) and classified the pixel to the land cover category with the higher likelihood.

The RF binary classification model was assessed in the pilot study regions using the validation points (ref. Section 3.6). When the classification overall accuracy was not satisfactory (less than 90%), the model was adjusted by refining the training polygons by excluding cultivated pasture polygons with possible mixed pixel problems until the overall accuracy was over

205 95%.

### 3.6 Accuracy assessment and area estimation

To assess the RF binary classification model's accuracy, we used 1,000 random independent validation points in the pilot study region of Qinghai Province as well as in the pilot study region of the Tibet Autonomous Region, as shown in Fig. 3b and 3d. Independent validation was used rather than cross-validation to avoid overestimating the accuracy of classified land cover

maps (Foody, 2002; Friedl et al., 2000). Two authors independently labeled the land cover type of each validation site as either cultivated pastures or not using high spatial resolution images on Google Earth. To visualize the spatial extent of a validation site, a 30-meter radius buffer circle with the validation site as the center was introduced. The authors used field knowledge gained during the 2021 campaign to label the validation sites. For instance, forage production companies in the pilot study regions typically managed cultivated pastures using heavy mechanical machines. As a result, tractor furrows were present, but

field ridges were not. In contrast, many field ridges were visible in regular croplands managed by small household farmers.

When both interpretations agreed, the identified land cover category was assigned to the validation point. In cases when the two interpretations did not agree, a third author was invited to resolve the conflict. The land cover category determined by the three coauthors was then assigned to the validation point. The labeled independent random validation reference points were illustrated in Fig. 3b and 3d and summarized in Table 3.

The overall, producer's, and user's accuracies of the trained RF binary classification model in the pilot study regions were calculated. Since this was a binary classification, the F1 spatial consistency score (based on precision and recall) was also calculated. The kappa coefficient was not reported since it has been proven unsuitable for assessing land cover maps' accuracy (Foody, 2020). To compute the uncertainties of the overall accuracy, producer's accuracy, user's accuracy, and F1 spatial consistency score, we used the method described in (Yang et al., 2024). In addition, we computed the areas of cultivated

pastures in each of the pilot study regions and the uncertainties using the method described in (Olofsson et al., 2014).





**Figure 4.** Overview of the method for mapping cultivated pastures on the Tibetan Plateau.

**Table 3.** Summary of the labeled independent random validation points in the pilot study regions in Qinghai Province and the Tibet Autonomous Region.

| Region | Land cover category | Number of points |
|---|---|---|
| | *Cultivated pasture* | 77 |
| Qinghai Province | *Other* | 923 |
| | Total | 1000 |





| | | |
|---|---|---|
| | *Cultivated pasture* | 16 |
| The Tibet Autonomous Region | *Other* | 984 |
| | Total | 1000 |

## 4 Results

### 4.1 The maps of cultivated pastures in the pilot study regions in 2021

Fig. 5a and 6a show the extent of cultivated pastures in the pilot study regions in 2021, when the field campaign was conducted. In the pilot region in Qinghai Province, cultivated pastures were mainly distributed around Qinghai Lake and in valleys with
favorable hydrothermal conditions. In the pilot study region in the Tibet Autonomous Region, cultivated pastures were primarily located in low-altitude regions such as Shigatse, Lhasa, Shannan, Nyingchi, and Chamdo. Fig. 5b and 6b show the number of valid Landsat OLI observations during the 2021 growing season in the pilot study regions. The RF binary classification model generated the likelihood of each pixel belonging to one of two land cover categories: *cultivated pasture* and *other*. The land cover category with the higher likelihood was assigned to the pixel grid. For instance, Fig. 5c and 6c
illustrate the likelihood of each pixel belonging to the *cultivated pasture* category.

The areas of cultivated pastures in the pilot study regions were estimated (Fig. 7a). In the pilot study region in Qinghai Province, cultivated pastures covered 0.422 ± 0.03 Mha, and in the pilot study region of the Tibet Autonomous Region, 0.058 ± 0.03 Mha. In addition, Fig. 5 and 6 provide close-ups illustrating the boundaries of cultivated pastures. The distinct boundaries between *cultivated pasture* and *other* suggested that our RF binary classification method could effectively identify
cultivated pastures.



**Figure 5.** Mapped cultivated pastures in the Qinghai Province pilot study region in 2021. (a) Despite displaying a similar spectrum to natural grasslands during the peak growing season, cultivated pastures were identified by the RF binary classification model. (b) The number of good observations from Landsat OLI in the pilot study region in 2021. Close-up views of the three regions A, B, and C are on the right panel. In particular, region B had fewer than 5 good observations, but this did not prevent the RF binary classification algorithm from identifying cultivated pastures there. (c) The likelihood of each 30-m grid being classified as *cultivated pasture* in the pilot study region. The likelihood was calculated by the RF binary classification algorithm.



**Figure 6.** Mapped cultivated pastures in pilot study region of the Tibet Autonomous Region in 2021. The panel descriptions are the same as those of Fig. 5.

## 4.2 Accuracy of the mapping method

The likelihood of cultivated pastures being classified as *cultivated pasture* was way higher than being classified as *other* in the pilot study regions of both Qinghai Province and the Tibet Autonomous Region (Fig. S2). As assessed by our independent random validation sites, the cultivated pasture map in Fig. 5a has an overall accuracy of 96.5% ± 0.5% and an F1 spatial consistency score of 80% ± 12%, the land cover map in Fig. 6a has an overall accuracy of 99.2% ± 0.3% and an F1 spatial consistency score of 85% ± 14% (Table 4). In the Qinghai Province pilot study region, the producer's accuracy (92.3% ± 2.9%) was higher than the user's accuracy (71.0% ± 4.6%) for cultivated pastures (Fig. 7b), indicating a higher commission error than omission error. In the pilot study region of the Tibet Autonomous Region, the user's accuracy (95.6% ± 3.1%) was higher than the producer's accuracy (88.2% ± 4.6%) for cultivated pastures (Fig. 7b), indicating a higher omission error than commission error. In the pilot study regions where the climates and landscapes differ substantially, the overall accuracies of

our cultivated pastures mapping method were both higher than 95%, indicating that we could use the mapping method to map cultivated pastures on the Tibetan Plateau.

We also compared the area of cultivated pastures mapped using remote sensing with those from the government statistics. In the government statistics reports on grasslands in 2021, there were 13 county-level summaries of the area of cultivated
pastures in Qinghai Province, and 12 county-level summaries of the area of cultivated pastures in the Tibet Autonomous Region (Table S1). We compared theses with the areas of cultivated pastures mapped using remote sensing data (Fig. 8). Our estimates using remote sensing data matched well with those from the government statistics.

**Table 4.** The error matrix for the cultivated pasture maps in the pilot study regions in Qinghai Province and the Tibet Autonomous Region.

|  | Category | *Cultivated pasture* | *Other* | User's accuracy |
|---|---|---|---|---|
|  | *Cultivated pasture* | 0.072 | 0.036 | 71.0% ± 4.6% |
| Qinghai Province | *Other* | 0.011 | 0.819 | 99.3% ± 0.3% |
|  | Producer's accuracy | 92.3% ± 2.9% | 96.8% ± 0.5% | 96.5% ± 0.5% |
|  | *Cultivated pasture* | 0.002 | 0.002 | 95.6% ± 3.1% |
| The Tibet Autonomous Region | *Other* | 0.001 | 0.995 | 99.4% ± 0.2% |
|  | Producer's accuracy | 88.2% ± 4.6% | 99.8 ± 0.2% | 99.2% ± 0.3% |

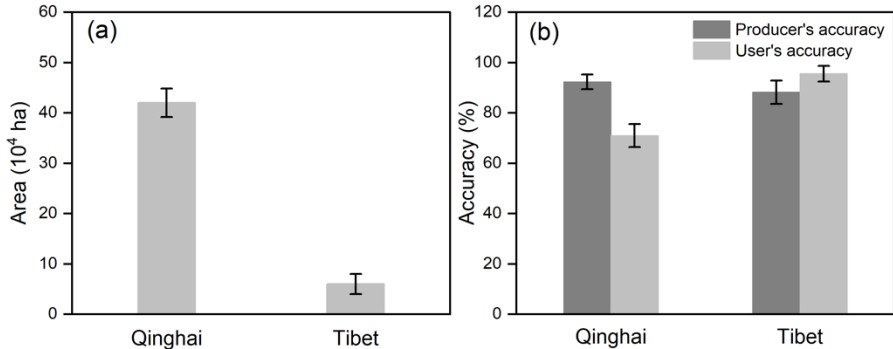

275
**Figure 7.** The areas and classification accuracies of the mapped cultivated pastures in the pilot study regions in Qinghai province and the Tibet autonomous region.

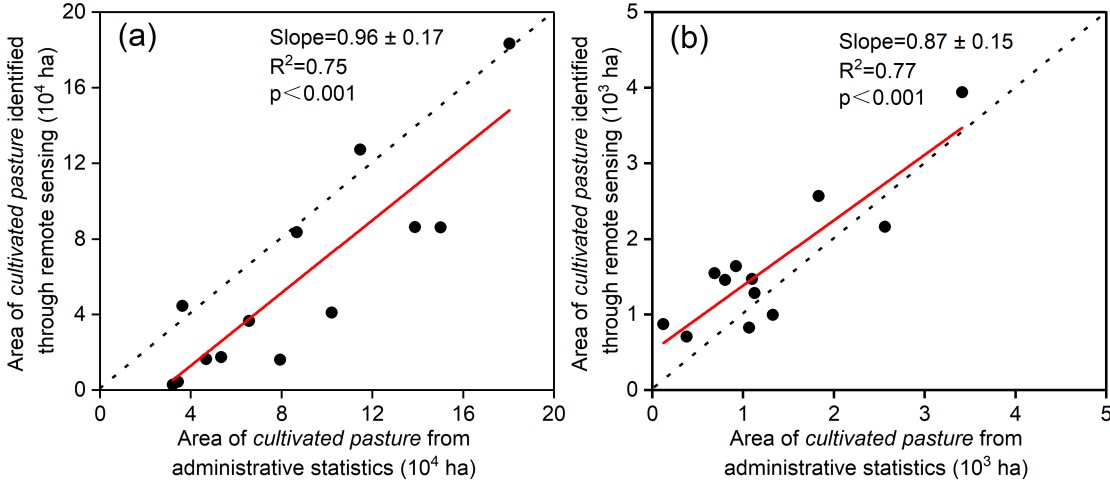

**Figure 8.** Comparison of the areas of cultivated pastures mapped using remote sensing with those from the government statistics at the county level for 2021 in (a) Qinghai Province and (b) the Tibet Autonomous Region. The government statistics of cultivated pasture areas were from Qinghai Forestry and Grassland Bureau and the Tibet Autonomous Region Agriculture and Rural Affairs Department.

### 4.3 The spatial and temporal distribution of cultivated pastures on the Tibetan Plateau

Using the Landsat data from 1988 to 2021 over the Tibetan Plateau, we mapped the annual distribution of cultivated pastures in Qinghai Province and the Tibet Autonomous Region (Fig. 9). Generally, cultivated pastures mainly appeared in certain regions on the plateau: (1) in Qinghai Province, the regions are the Qinghai Lake area (the counties of Gangcha, Haiyan, Huangyuan, Gonghe, Guide, Dulan, and Wulan), the Qilian Mountain area (the counties of Qilian and Menyuan), the Three-River Headwaters area (the counties of Guinan, Tongde, Xinghai, Maqin, Dari, and Jiuzhi), as well as the Yushu area; (2) in the Tibet Autonomous Region, the regions are northern Tibet (the counties of Bange, Nima, Dinqing, Naqu, and Gaize), southeastern Tibet (the counties of Longzi, Qunar, and Basu), and the watersheds of the rivers Yarlung Tsangpo, Lhasa, and Nianchu (the counties of Gongga, Linzhou, Dangxiong, Kangma, Nanmuling, and Saga). The cultivated pastures in Qinghai were more clustered, while in Tibet they were more dispersed.

Cultivated pastures in Qinghai Province existed longer than in the Tibet Autonomous Region (Fig. 10). In Qinghai Province, many cultivated pastures existed for more than 20 years from 1988 to 2021, especially around the Qinghai Lake. In the Tibet Autonomous Region, cultivated pastures were generally in existence for less than 10 years.

There were government statistics data for cultivated pasture areas at the province level in Qinghai Province and the Tibet Autonomous Region from 2001 to 2021. But only in the years from 2010 to 2017 did the government's statistical caliber of cultivated pastures align with our definition of cultivated pastures, and the cultivated pasture areas reported by government statistics were reasonably close to those mapped using remote sensing from 2010 to 2017 (Fig. 11).

The average area of cultivated pastures on the plateau from 1988 to 2021 was approximately 1.0 Mha, with ~ 0.7 Mha in Qinghai and ~ 0.3 Mha in Tibet. From 1988 to 2021, there had been an increasing trend in the area of cultivated pastures on the plateau based on our mapping results using remote sensing data (Fig. 11a), and the increasing trend was mostly due to the

expansion of cultivated pastures in Qinghai (Fig. 11b and 11c). The increasing rate of the area of cultivated pastures on the Tibetan Plateau was 33.5 ± 2.5 Kha per year. In most counties of Qinghai Province and the Tibet Autonomous Region, cultivated pasture areas did not change much from 1988 to 2021 (Fig. 12). Cultivated pastures expanded substantially in Gonghe, Gangcha, Guinan, and Xinghai, while contracted notably in Karuo and Luolong (Table S2).

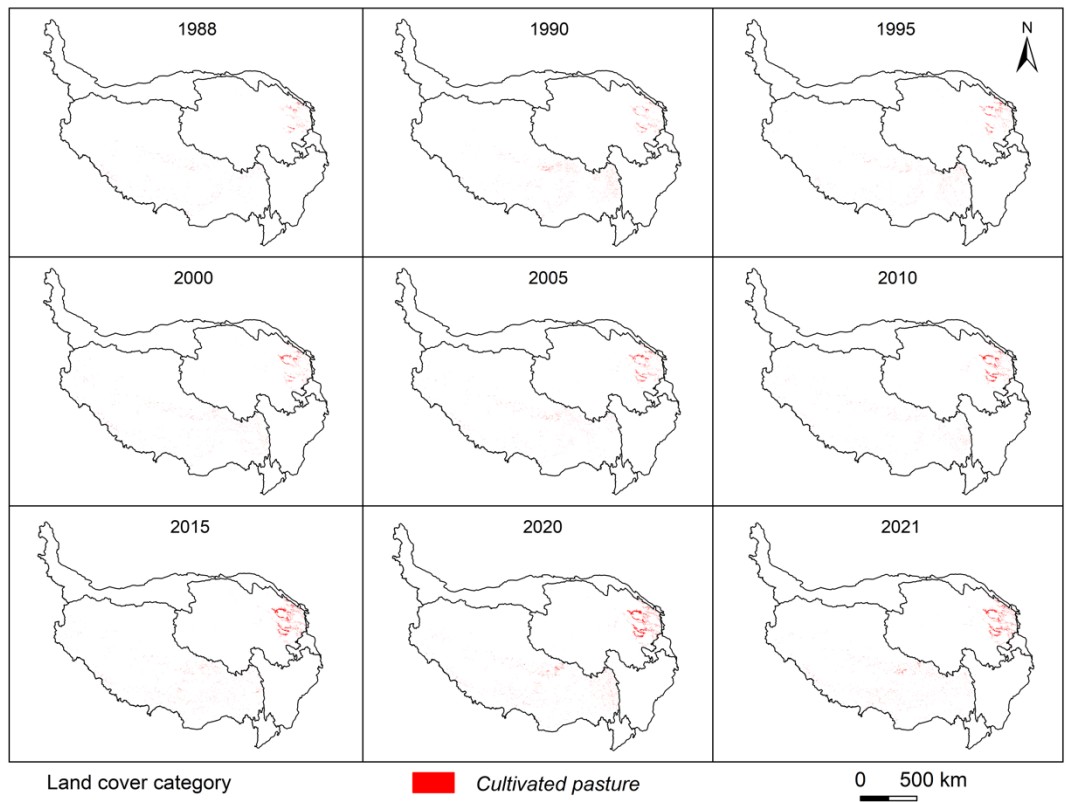

**Figure 9.** The maps of cultivated pastures in Qinghai Province and the Tibet Autonomous Region on the Tibetan Plateau from 1988 to 2021 (selected years are displayed for brevity, and the whole time series can be found in Fig. S3 of the supplementary material).

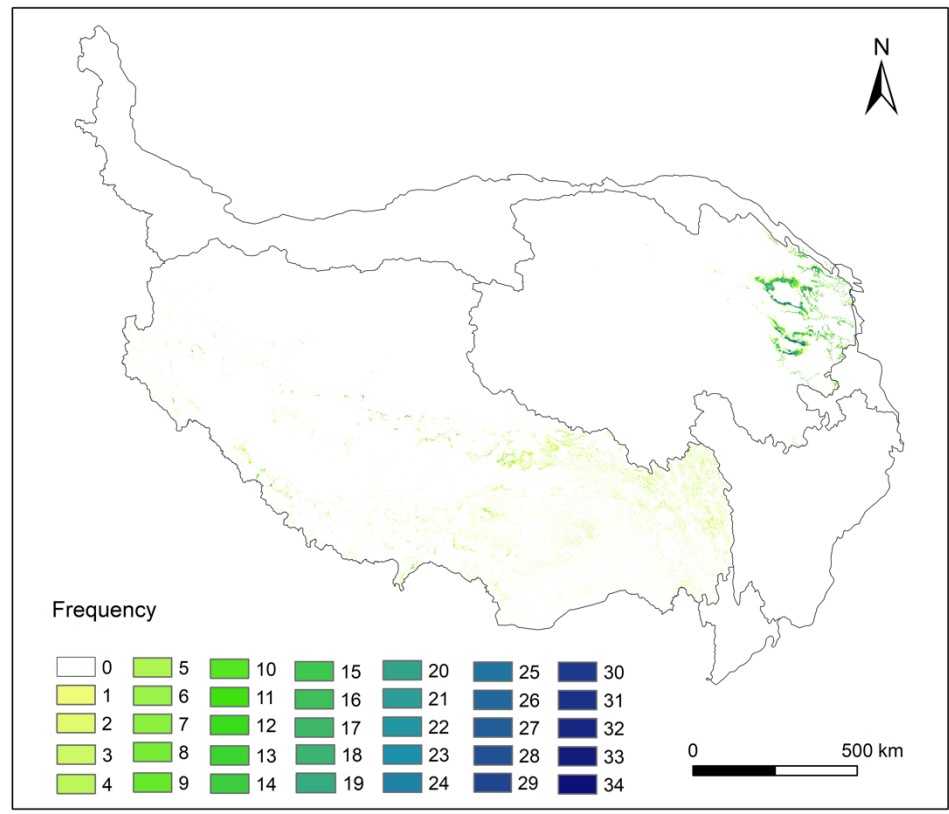


**Figure 10.** The number of years that *cultivated pasture* existed in each 30-m grid in Qinghai Province and the Tibet Autonomous Region from 1988 to 2021.


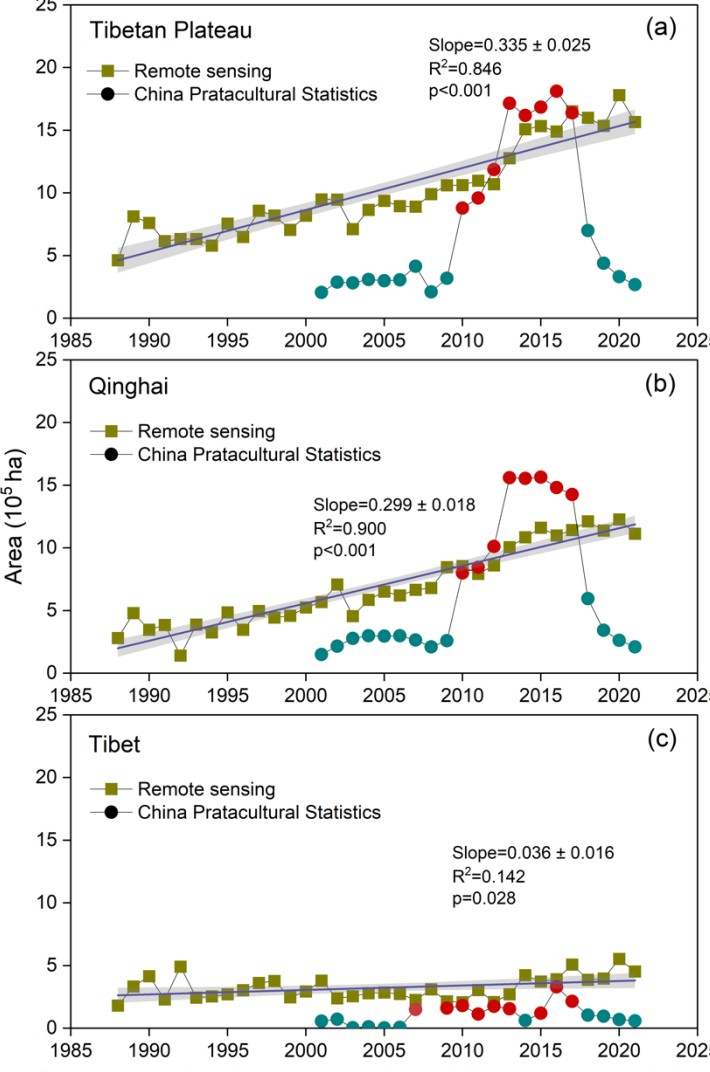

**Figure 11.** The time series of cultivated pasture areas based on remote sensing and the government statistics data for (a) the Tibetan Plateau, (b) Qinghai Province, and (c) the Tibet Autonomous Region. The red dots in the China Pratacultural Statistics time series correspond to the years when the cultivated pasture statistical caliber aligned with our definition of cultivated pasture in the remote sensing mapping efforts.

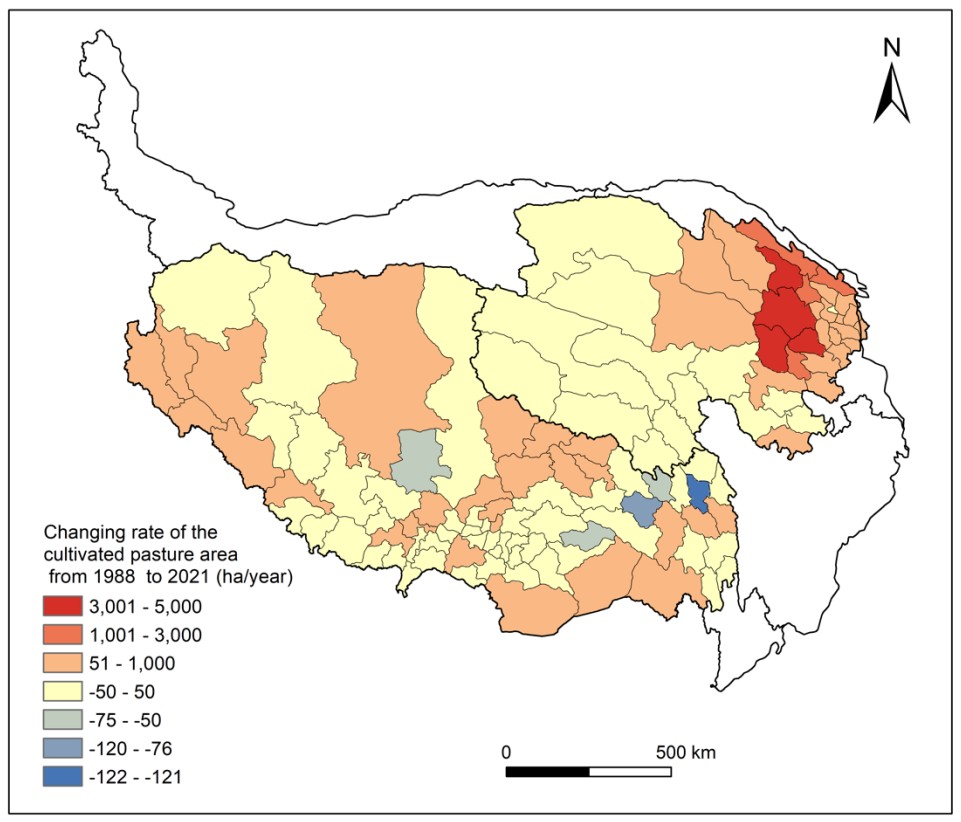

**Figure 12.** Trend of cultivated pasture areas from 1988 to 2021in each county of Qinghai Province and the Tibet Autonomous Region. Cultivated pastures expanded substantially in Gonghe, Gangcha, Guinan, and Xinghai, while contracted notably in Karuo and Luolong.

## 5 Discussion

In this study, with decent accuracy, we successfully mapped the distribution of cultivated pastures on the Tibetan Plateau for

the first time. Compared to previous efforts aimed at mapping only certain types of cultivated pastures (Parente et al., 2019;

Wang et al., 2022; Yang et al., 2021), this method could map general cultivated pastures.

### 5.1 The method for cultivated pasture mapping

A distinctive feature of our mapping method is using the spectral-temporal metrics of remote sensing data. This contrasts with

previous methods of using the time series of all valid remote sensing observations during the growing season (McInnes et al.,

2015; Parente et al., 2019; Yang et al., 2021). Our method can implicitly capture important phenological information during

the growing season with calculated statistics of the time series of all valid remote sensing observations in the growing season,

resulting in accurate identification of cultivated pastures on the Tibetan Plateau.

Our mapping method also owes its success to the large volume of training data from various types of cultivated pastures

recorded during the extensive fieldwork. This allowed the binary classification model to incorporate the spectral-temporal





information of general cultivated pastures. Unlike previous mapping methods (Fisher et al., 2018; Parente et al., 2019; Wang

et al., 2022; Yang et al., 2021), our training data came from training polygons instead of training points, resulting in an increased volume of training data.

In addition, we only used satellite remote sensing data from one year instead of multiple years (Potapov et al. 2022), because inter-annual crop rotation exists on cultivated pastures on the Tibetan Plateau. Although multi-year composite data can help to mitigate atmospheric disturbances such as clouds and cloud shadows, they can also lead to misclassification issues

due to the rotation of cultivated pastures.

This study employed an independent validation method (Yang et al., 2021) to assess the classification accuracy, instead of the cross-validation method used in some previous studies (Ashourloo et al., 2018; Wang et al., 2022). Cross-validation may overestimate classification accuracy (Friedl et al., 2000; Foody, 2002) because the model is trained and validated using the same set of reference data, with certain types of cultivated pastures not included in the reference dataset.

**5.2 Accuracy assessment of mapped cultivated pastures**

The accuracy assessment of the binary classification model is essential in the cultivated pasture mapping practice (Olofsson et al. 2014; Stehman et al. 2019). In this study, the RF binary classification model demonstrated a good ability to identify cultivated pastures, with an overall accuracy of 97.05% ± 0.4% and an F1 spatial consistency score of 0.83 ± 0.14. This was superior to a recent study mapping the spatial extents of green fodder lands in the northeastern Tibetan Plateau using Landsat

data with overall accuracies of 94.2%, 93.1%, and 96.6% in 2010, 2015, and 2019 (Yang et al., 2021). The cultivated pasture mapping accuracy obtained in this study (overall accuracy of 97.05% ± 0.4%) was close to that reported by Wang et al. (2022). Wang et al. (2022) solely mapped oat pastures at the county scale in Shandan Racecourse in the northeastern Tibetan Plateau with an overall accuracy of 98%. Our results were also better than another effort mapping native and non-native grasslands using MODIS NDVI time series data in Canada, which achieved an overall accuracy of 73% (McInnes et al., 2015).

We found that the differences in the spatial fragmentation of cultivated pastures in Qinghai Province and the Tibet Autonomous Region could affect the accuracy of the cultivated pasture maps. During our field visits, we noticed that the spatial distribution of cultivated pastures in the Tibet Autonomous Region was much more fragmented and dispersed than in Qinghai Province. In Qinghai Province, the cultivated pastures were sometimes in the shape of long stripes and next to regular croplands, which might cause the mixed pixel problem and lower the identification accuracy of cultivated pastures (user's accuracy of

71.0% ± 4.6% in Qinghai vs. user's accuracy of 95.6% ± 3.1% in Tibet), since the spectral characteristics of regular croplands and cultivated pastures are very similar during the peak growing season (Yang et al., 2021; Wang et al., 2022).

**5.3 Comparison of the mapped cultivated pasture areas with government statistics**

We found that the areas of cultivated pastures identified through remote sensing were comparable to the areas of cultivated pastures reported in government statistics for Qinghai and Tibet (e.g. Li and Wang, 2017). For example, in 2021, the year for

which we trained the RF binary classification model for cultivated pastures on the Tibetan Plateau, we mapped 1.57 Mha





cultivated pastures in Qinghai and Tibet, while the area of cultivated pastures in 2017 reported in the government statistics was 1.640 Mha. We used the statistics data of cultivated pastures in 2017 because it was the closest year in which the statical caliber used by the annual *China Pratacultural Statistics* agreed with our definition of cultivated pastures.

The time series of the area of cultivated pastures on the Tibetan Plateau mapped through remote sensing from 1988 to 2021 was likely driven by the implementation of ecological and agricultural policies (Fig. S4; Schils et al. 2022; Zhou et al. 2020). However, the time series of the area of cultivated pastures on the Tibetan Plateau reported in the annual *China Pratacultural Statistics* did not exhibit any correlation with the implementation of ecological and agricultural policies, and was severely distorted by the shift of the statistical caliber of cultivated pastures in it. The areas of cultivated pastures reported in the annual *China Pratacultural Statistics* for the years from 2010 to 2017 were substantially higher than those for the remaining years

(Fig. 9); nevertheless, they were close to the remote sensing estimates in the period from 2010 to 2017. The time series of the areas of cultivated pastures mapped by remote sensing from 1988 to 2021 on the Tibetan Plateau was more realistic than the government statistics data.

The increasing trend in the area of cultivated pastures estimated through remote sensing in this study agrees with a previous relevant regional study of cultivated pasture mapping on the Tibetan Plateau. Yang et al. (2021) found a rapid expansion of

green fodder lands in the northeastern Tibetan Plateau from 1.63 Kha in 2010 to 13.61 Kha in 2019.

### 5.4 Limitations and future prospects

The time series of cultivated pasture maps on the Tibetan Plateau were produced for the first time; nevertheless, the mapping method and the maps had several limitations and could be improved in future.

(1) Some long and narrow cultivated pastures might not be mapped on the 30-meter cultivated pasture map. On the Tibetan

Plateau, especially in Qinghai Province, cultivated pastures were sometimes in the shape of long and narrow (less than 30 m) stripes and adjacent to regular croplands, causing the mixed pixel problem. When the proportion of cultivated pastures was less than 50% in the mixed pixels, the mixed pixels tended to be classified as regular croplands, causing long and narrow stipes of cultivated pastures not to be mapped. The mixed pixel problem could be alleviated using higher spatial resolution remote sensing data starting in the near past, such as Sentinel-2 data starting in 2015 (Phiri et al., 2020). We did not use Sentinel-2

data because we were interested in the long-term spatial distribution of cultivated pastures on the Tibetan Plateau.

(2) Although the cultivated pasture map was validated in two pilot study regions with different climates, landscapes, soil properties, and ecological conditions for 2021, and although the areas of cultivated pastures matched well with government statistics at both the county and province levels, the cultivated pasture maps from 1988 to 2021 could be further validated by other researchers or practitioners in regional or local applications. The cultivated pasture mapping method could be improved

with their usage feedback.





## 6 Data Availability

The cultivated pasture maps generated in this study can be accessed at https://doi.org/10.5281/zenodo.14271782 (Han et al., 2024). All maps are at the 30 m (~ 0.00027°) spatial resolution under the EPSG:4326 (WGS84) spatial reference system.

## 7 Conclusions

Cultivated pastures are crucial forage sources for livestock on the Tibetan Plateau. Additionally, they have significant implications for the region's ecological conservation and restoration efforts. In this study, we mapped cultivated pastures from 1988 to 2021 on the Tibetan Plateau using satellite remote sensing data for the first time. The mapping method performed satisfactorily with an overall accuracy of 97.05% ± 0.4% and an F1 spatial consistency score of 82.51% ± 14.22% (Precision: 90.04% ± 6.18%, Recall: 76.74% ± 9.91%). At both the province and county levels, the cultivated pasture area estimated in this study matched well with government statistics. The area of cultivated pastures on the Tibetan Plateau experienced a notable increasing trend from 1988 to 2021, at a rate of 33.5 ± 2.5 Kha per year. The cultivated pasture mapping method can be adopted to identify cultivated grasslands in other regions of the world.

**Supplement.** The supplement related to this article is available online at: https://doi.org/10.5xxxx/essd-xxxxx-supplement.

**Author Contributions.** J-SH and JB designed the research. BH, TY, YT, and JB performed the analysis. BH and MG collected the field data and remote sensing data. BH, JB, and J-SH wrote the first draft of the manuscript. All authors participated in the review and editing of the manuscript.

**Competing Interests.** The contact author has declared that none of the authors has any competing interests.

**Disclaimer.** Publisher's note: Copernicus Publications remains neutral with regard to jurisdictional claims in published maps and institutional affiliations.

**Acknowledgements.** We gratefully acknowledge all data providers whose data have been used in this study and would like to thank the topical editor and the anonymous referees for their constructive comments.

**Financial Support.** This study was supported by the National Natural Science Foundation of China (Grant no. 32192461 and 32130065) and the Consulting Project of the Chinese Academy of Engineering (Grant no. 2023-XY-28, 2024-XZ-56, and GS2023ZDI01).

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
