# Peer review of "An annual 30 m cultivated pasture dataset of the Tibetan Plateau from 1988 to 2021"

_Earth System Science Data, 2024_

## Author Comment (AC1)

**Response to Reviewer 1 Comments**

I enjoyed this paper. The knowledge gap is well established, the methodology is solid, the results and discussion address the study objectives. I have a few comments for the authors' consideration.

Response: We sincerely thank the reviewer for the positive feedback and for acknowledging the strength of our study. We are pleased to know that the knowledge gap is clearly established, the methodology is considered robust, and the results and discussion effectively address the study objectives. We have carefully considered the reviewer's comments and have provided responses and revisions as outlined below.

Specific Comments:
Since this study focuses specifically on Qinghai and Tibet (not the entire Tibetan Plateau), I recommend adjusting the title to more accurately reflect this scope. Additionally, I suggest adding a boundary outlining these two regions in Figure 1 to clearly define the study area. For Figures 9 and 10, a gray shadow as the background could enhance the visibility of the study area and improve contrast, as the current color is hard to discern. Adding the boundary and shaded areas will help readers understand the study area more effectively and avoid potential misinterpretations. For example, based on Figures 9 and 10, I initially concluded that there is no cultivated pasture in Xinjiang, Gansu, and Sichuan, which could be misleading.

Response: We sincerely thank the reviewer for the thoughtful and constructive suggestions regarding the clarity of the study area and figure presentation. In response:
- We have added a boundary outlining Qinghai Province and the Tibet Autonomous Region in Figure 1 to more clearly delineate the geographic scope of the study.
- To improve visual contrast and enhance the visibility of the study area in Figures 9 and 10, we have incorporated a gray shaded background. These revisions are intended to help readers more effectively interpret the spatial context and avoid potential misinterpretations—such as assuming the absence of cultivated pastures in surrounding regions like Xinjiang, Gansu, or Sichuan.

Regarding the title of the manuscript, we acknowledge the reviewer's point and agree that the study is geographically limited to Qinghai and Tibet (in order to match government statistics), which together comprise approximately 77% of the Tibetan Plateau. For the sake of brevity and consistency, we have retained the current title, but we have made clarifications in the manuscript text to clearly define the spatial extent of the study at the outset.

The revised Figures 1, 9, and 10 have been updated accordingly and are included in the revised manuscript. We greatly appreciate the reviewer's input, which has helped improve the clarity and interpretability of our work. The revised Figures 1, 9, and 10 are as follows.

[Figure]

**Figure 1.** The land cover types of the study region and the distribution of the pilot study regions in Qinghai and Tibet. The land cover data source: the Resource and Environment Science and Data Center (*http://www.resdc.cn/*) of the Chinese Academy of Sciences. The binary classification model for mapping cultivated pastures was trained and validated in the pilot study regions.

[Figure]

**Figure 9.** The maps of cultivated pastures in Qinghai Province and the Tibet Autonomous Region on the Tibetan Plateau from 1988 to 2021 (selected years are displayed for brevity, and the whole time series can be found in Fig. S3 of the supplementary material).

[Figure]

**Figure 10.** The number of years that cultivated pasture existed in each 30-m grid in Qinghai Province and the Tibet Autonomous Region from 1988 to 2021.

The Methods section could be made clearer with the following revisions:
The growing season is first introduced in Section 3.1, while the description of quantile extraction appears in Section 3.2. I suggest merging these two sections to ensure a smoother logical flow. For instance, begin with an introduction to SR and the seven spectral indices, followed by an explanation of the satellite products and the growing season sampling process, and conclude with the topography data. Feel free to disregard this suggestion if it doesn't fit the structure of the paper.

Response: We sincerely thank the reviewer for this thoughtful and detailed suggestion. We fully understand the rationale behind the proposed restructuring and agree that such an approach could enhance the logical flow of the Methods section in certain contexts. However, after careful consideration, we believe that keeping Sections 3.1 and 3.2 separate allows us to present the remote sensing metrics and the sampling strategy in a more focused and accessible manner. In our view, this structure better supports readers who may wish to reference specific methodological components independently.

The use of quantiles is not entirely clear. Were they used as separate model inputs, or did they interact in some way? If they were used individually, the importance of each

quantile likely varies for different pixels. Clarifying this would improve understanding.

Response: We thank the reviewer for this valuable comment and appreciate the opportunity to clarify this point. We apologize for any confusion caused in the original text. In our study, the quantiles were used as independent input variables in the Random Forest classification model and did not interact with one another. As is standard with Random Forest models, the importance of each input variable is determined during the training process and is subsequently applied uniformly across all pixels in the classification. In other words, the model assigns a fixed importance to each quantile metric based on its contribution to the overall classification accuracy, rather than varying by pixel.
To improve clarity, we have revised the relevant section in the manuscript and have highlighted the changes using the "Track Changes" feature in Word. We thank the reviewer once again for helping us enhance the transparency and precision of our methodological description.

Several types of cultivated pasture were used as training data. Since the model is binary (cultivated vs. other), how were these different types handled in the model? Were they treated as equivalent to cultivated pasture, or did the model account for their distinctions? Additionally, how did the model perform across these various types?

Response: We thank the reviewer for this thoughtful question. As noted, several types of cultivated pastures were included in the training dataset. In our classification framework, these different types were treated uniformly under the general land cover category of "cultivated pastures." This approach aligns with the objective of the study, which was to map the overall spatial distribution of generalized cultivated grasslands across Qinghai and Tibet, rather than to distinguish between specific subtypes of cultivated pastures.
Accordingly, the Random Forest model was trained to identify the broader category without differentiating between its internal variations. While we acknowledge that model performance may vary across specific types of cultivated pastures, evaluating performance at that level of granularity was beyond the scope of the present study. We appreciate the reviewer's suggestion, and we agree that this could be a valuable direction for future research.

The performance of the model should be presented more explicitly, particularly regarding the importance of different input drivers.

Response: We thank the reviewer for the constructive and insightful comment. In response, the following text has been added to Section 4.2 to more explicitly present

the performance of the model, with particular emphasis on the relative importance of the different input variables. These additions are intended to enhance the clarity and transparency of our methodology and results. The corresponding revisions have been highlighted in the manuscript using the "Track Changes" feature in Word.

*"The importance rankings of input variables in the trained Random Forest models for classifying cultivated pastures revealed consistent patterns across Qinghai and Tibet (Table 5). In both regions, elevation emerged as the most influential variable, contributing 30.1% and 28.4% of the model importance in Qinghai and Tibet, respectively. Vegetation indices such as NDVI, EVI, NDWI, and NDPI also played major roles, collectively accounting for a substantial portion of the variable importance in both regions. For instance, NDVI contributed 14.7% in Qinghai and 18.2% in Tibet. Spectral bands (e.g., B2, B3, B4, B5) had moderate to low importance, while topographic variables such as slope and aspect, along with certain indices like NDBI and MNDWI, showed relatively minor contributions. These findings underscore the critical role of both topography and vegetation dynamics in distinguishing cultivated pastures on the Tibetan Plateau."*

**Table 5.** The importance of each input variable in the trained Random Forest models for classifying cultivated pastures in Qinghai and Tibet.

| | Index | Importance | Index | Importance | Index | Importance | Index | Importance |
|---|---|---|---|---|---|---|---|---|
| Qinghai | Elevation | 30.1% | B3 | 8.3% | B7 | 0.8% | Aspect | 0.1% |
| | NDVI | 14.7% | B5 | 5.4% | B4 | 0.6% | B1 | 0.1% |
| | EVI | 12.0% | NBR | 3.1% | NDBI | 0.4% | | |
| | NDWI | 10.6% | MNDWI | 2.8% | B6 | 0.4% | | |
| | NDPI | 9.1% | B2 | 1.2% | Slope | 0.3% | | |
| Tibet | Elevation | 28.4% | B2 | 7.6% | NDBI | 1.1% | B7 | 0.2% |
| | NDVI | 18.2% | B4 | 4.8% | Slope | 0.5% | Aspect | 0.1% |
| | EVI | 12.3% | B3 | 3.3% | B6 | 0.5% | | |
| | NDPI | 9.8% | B5 | 2.6% | B1 | 0.3% | | |
| | NDWI | 8.3% | NBR | 1.8% | MNDWI | 0.2% | | |

The field records used to train the model cover only a portion of the study area. Would it be feasible to extend the training data by using high-resolution satellite images for non-pilot regions?

Response: We thank the reviewer for this thoughtful comment. It is true that the training data were derived from field records collected in selected portions of the study area. While the use of high-resolution satellite imagery to supplement training data in non-pilot regions is a valuable suggestion, in our case, it was not feasible to reliably distinguish cultivated pastures from natural grasslands and conventional croplands using such imagery alone. The spectral and spatial characteristics of these land cover

types can be highly similar. Therefore, field-based validation was essential to ensure the accuracy and consistency of the training data. We appreciate the reviewer's suggestion and agree that exploring complementary methods to expand training data coverage could be a worthwhile direction for future research.

Spectral and topographic data alone may not be sufficient to accurately predict cultivated pasture, especially over time. I suggest considering additional drivers such as climate variables, soil properties, and human or livestock populations in the modeling process.

Response: We sincerely thank the reviewer for this insightful suggestion. We agree that incorporating additional drivers such as climate variables, soil properties, and human or livestock population data can offer valuable contextual information for land cover modeling. However, in this study, our modeling approach primarily relied on spectral and topographic data with a spatial resolution of 30 meters, while the suggested auxiliary datasets are typically available at much coarser spatial resolutions—often around 1 kilometer. This substantial mismatch in spatial resolution poses challenges for integration, particularly when mapping features as spatially heterogeneous as cultivated pastures.

Moreover, it is important to note that spectral data inherently reflect the influence of various environmental and anthropogenic factors, including climate conditions, soil characteristics, and land use practices. As such, much of the relevant information from these drivers is indirectly captured through spectral-temporal signatures.

Nevertheless, we appreciate the reviewer's recommendation, and we agree that incorporating additional drivers—where high-resolution data are available—could be a valuable direction for future refinement of the model.

The predicted area in Figure 8(a) appears to be consistently smaller than government statistics. Figure 11(b) also shows that the prediction for Qinghai is under-estimated. Are these discrepancies related to the limitations mentioned in point 1? They need to be clarified.

Response: We sincerely thank the reviewer for the thoughtful and constructive comments, which prompted us to further reflect on the discrepancies observed and to strengthen our explanation in the manuscript.

**Regarding Figure 8(a):**
The predicted area of cultivated pastures in our results appears smaller than the corresponding government statistics. This discrepancy is primarily due to differences in classification criteria: the definition of cultivated pastures used in our study is not fully aligned with that employed by local government agencies. As a result, we focused our comparison on correlation metrics at the county level, rather than on absolute or relative differences. The coefficients of determination ($R^2$) were 0.75 for Qinghai and 0.77 for Tibet, indicating a strong spatial agreement and reinforcing the

reliability of our dataset. We have revised the manuscript accordingly to include the following statement:

*"Since the statistical criteria for cultivated pastures used by local governments do not fully align with our definition, we focused the comparison between our results and government statistics at the county level on correlation metrics rather than absolute or relative errors (Fig. 8 and Table S1). The coefficients of determination were 0.75 for Qinghai and 0.77 for Tibet, indicating the reliability of our results."*

**Regarding Figure 11(b):**

Upon closer examination of Figure 11(b), we observed a marked increase in the reported area of cultivated pastures in Qinghai in 2013 based on government statistics. This sudden change likely reflects a revision in the statistical criteria or reporting practices rather than an actual land cover change. Given this uncertainty, and the lack of transparency in government data methodologies, we opted not to perform a quantitative comparison between our results and the official figures. We are grateful to the reviewer for raising this point, which helped us refine our argument. The following clarification has been added to the manuscript:

*"A sharp increase in the area of cultivated pastures for Qinghai is reported in the 2013 government statistics (Fig. 11b), suggesting a potential shift in the statistical criteria for cultivated pastures that year. In contrast, our results show a more gradual increase, which likely reflects a more consistent and accurate representation of the actual expansion of cultivated pastures on the Plateau. These findings suggest that government statistics may warrant further scrutiny in future policy development related to cultivated pastures."*

Technical Comments:

Figure 3: For better visualization, use two distinguishable colors for the two categories. This will improve clarity and contrast.

Response: We thank the reviewer for the helpful suggestion to improve the visual clarity of Figure 3. In response, we have updated the figure using two more distinguishable colors—red and green—for the two categories. This adjustment enhances contrast and improves the overall readability of the map. The revised Figure 3 is provided below.

[Figure]

| | Cultivated pasture | | Other |

**Figure 3.** The spatial distribution of the training polygons and the validation points in the pilot study regions. The training polygons (a, c) were recorded during the 2021 field campaign, and the 1,000 independent random validation points (b, d) in each pilot study region were labelled with the aid of high-resolution images on Google Earth.

---

## Author Comment (AC2)

**Response to Reviewer 2 Comments**

General Comments:

Cultivated pastures maps are essential for effective grassland resource management. This study uses a Random Forest (RF) binary classification model to identify cultivated pastures distribution on the Tibetan Plateau and distinguish between artificial grasslands, natural grasslands, and croplands. The authors have produced annual distribution maps of cultivated pastures in Qinghai Province and the Tibet Autonomous Region from 1988 to 2021. I personally think this research is significant but challenging. After carefully reading this paper, I have serious concerns about the credibility of its data.

Response: We sincerely thank the reviewer for taking the time to thoroughly evaluate our manuscript. We are grateful for the recognition of both the significance and complexity of our research. We fully acknowledge the concerns raised regarding the credibility of the data and appreciate the opportunity to clarify and strengthen these aspects. In response, we have carefully addressed each of the specific comments, and the corresponding revisions have been incorporated into the manuscript. Please find our detailed responses below.

Major concerns:

Line24: The abstract states that there are 2000 independent validation samples, but in Figure 4 and line 207, the number is given as 1000. The authors should check this discrepancy. Moreover, among these 2000 independent validation samples, only 93 are for cultivated pastures, which is clearly too small.

Response: We thank the reviewer for this insightful comment and apologize for any confusion caused by the inconsistency in the description of the validation sample size. To clarify, we used 1,000 randomly selected independent validation points in the pilot study region of Qinghai Province, and another 1,000 in the pilot study region of the Tibet Autonomous Region, resulting in a total of 2,000 validation points. We have revised the sentence in line 207 to reflect this more clearly:

*"To evaluate the accuracy of the cultivated pasture mapping, we used 1,000 randomly selected independent validation points in the pilot study region of Qinghai Province and another 1,000 in the pilot study region of the Tibet Autonomous Region, as shown in Figures 3b and 3d."*

Regarding the number of validation points specifically for cultivated pastures (93 in total), we acknowledge that this is relatively small. However, our evaluation indicates that this number was sufficient to yield reliable and statistically robust accuracy estimates. The resulting overall accuracy was 96.5% ± 0.5% for Qinghai and 99.2% ± 0.3% for Tibet, with relatively low uncertainty. These estimates were calculated using the method proposed by Yang et al. (2024), which enables the computation of confidence intervals for accuracy metrics such as overall accuracy, producer's accuracy, user's accuracy, and the F1 spatial consistency score. As stated in lines 223–224:

*"To compute the uncertainties of the overall accuracy, producer's accuracy, user's accuracy, and F1 spatial consistency score, we used the method described in Yang et al. (2024)."*

This method ensures that the sample size is statistically sufficient for the intended accuracy assessment. If the number of validation samples had been inadequate, we would have observed considerably greater uncertainty in the resulting estimates. We appreciate the reviewer's comment, which allowed us to clarify this important aspect of the study.

Line160~230: a) The authors mention in Lines 160-163 that they surveyed 81 cultivated pastures in Qinghai Province and 114 samples in Tibet. From Table 2, it appears that all field survey samples were used for model training. So, the origin of the 93 independent validation samples for cultivated pastures in Table 3 is unclear and needs to be verified by the authors. b) The proportion of other types of data is 91% in the training sample and even higher, at 95%, in the validation data, while the proportion of cultivated pastures is very low. This distribution is unreasonable and may affect the accuracy and reliability of the model. c) In machine learning studies similar to this study, it is common to use 70% of the data for training and 30% for validation. However, in this study, the training sample size is 2089 and the independent sample size is 2000. The authors should explain why they chose this proportion to divide the training and validation datasets.

Response: We appreciate the reviewer's comment. We have addressed the three concerns as follows.

a) The 93 independent validation samples for cultivated pastures among the 2,000 independent random validation samples were labeled by two co-authors. As described in Section 3.6, *Accuracy Assessment and Area Estimation*: "Two authors independently labeled the land cover type of each validation site as either cultivated pastures or not using high spatial resolution images from Google Earth" (lines 210–211).

b) Yes, the number of validation points for other land cover types was significantly larger than that for cultivated pastures. This discrepancy arises because validation points were randomly generated within the pilot study regions, where other land cover types dominate. Nonetheless, although the number of validation samples for cultivated pastures is relatively small, it is sufficient to provide a reliable accuracy estimate for the cultivated grassland map with small uncertainty—96.5% ± 0.5% for Qinghai and 99.2% ± 0.3% for Tibet. The required number of validation points to achieve statistically valid accuracy estimates can be determined using the method proposed by Yang et al. (2024), which we applied in this study. Our findings indicate that, under this validation scenario, a large number of validation samples was not necessary to obtain robust accuracy estimates.

c) In contrast to the reviewer's suggestion and many previous studies, we did not use cross-validation. This decision was based on findings from two seminal studies on land cover classification accuracy assessment, which highlight that cross-validation

can lead to an overestimation of classification accuracy.

Foody, Giles M. 2002. "Status of Land Cover Classification Accuracy Assessment." Remote Sensing of Environment 80, no. 1: 185–201. https://doi.org/10.1016/s0034-4257(01)00295-4.

Friedl, M. A., C. Woodcock, S. Gopal, D. Muchoney, A. H. Strahler, and C. Barker-Schaaf. 2000. "A Note on Procedures Used for Accuracy Assessment in Land Cover Maps Derived from AVHRR Data." International Journal of Remote Sensing 21, no. 5: 1073–77. https://doi.org/10.1080/014311600210434.

Line195-205: I agree with the author's description in the introduction section that cultivated pastures differ from croplands in their shorter duration of use. However, the study failed to incorporate this feature as a criterion for distinguishing cultivated pastures from croplands when building the model. As a result, the study's extraction of cultivated pastures cannot be clearly differentiated from croplands.

Response: We thank the reviewer for this insightful comment. We agree that the duration of green phenology is an important characteristic that can help differentiate cultivated pastures from conventional croplands. While our model does not explicitly incorporate the duration of phenology as a separate criterion, it captures key phenological patterns through statistical metrics derived from all valid remote sensing observations during the growing season. These spectral-temporal metrics implicitly reflect differences in vegetation growth dynamics, including variations in growing season length and intensity, which are often associated with land cover type.

This approach has proven effective in distinguishing cultivated pastures from croplands on the Tibetan Plateau. The classification results were validated with an overall accuracy of 97.05% ± 0.4%, along with an F1 spatial consistency score of 82.51% ± 14.22% (Precision: 90.04% ± 6.18%; Recall: 76.74% ± 9.91%). These results demonstrate that the model was able to capture the distinct characteristics of cultivated pastures, even without explicitly modeling the duration of phenology. We appreciate the reviewer's suggestion, and we agree that incorporating spectral-temporal features more directly could be a valuable avenue for future research.

Line250: a) As the images in Figure 5/6 show, it is difficult to distinguish cultivated pastures from croplands. So, what were the criteria for distinguishing between croplands and cultivated pastures when the authors did visual interpretation? b) There are only 34 datasets from 1988 to 2021. So, why does the maximum value in Figure 5/6b exceed 34?

Response: We sincerely thank the reviewer for the thoughtful and detailed comment. We have addressed the two concerns as follows:

a)  We acknowledge the challenge of visually distinguishing cultivated pastures from croplands in Figures 5 and 6 based solely on satellite imagery. To improve the reliability of our interpretation, we conducted extensive field campaigns in the regions illustrated in these figures, which provided critical ground-truth information and a comprehensive understanding of the local land use types. These field observations played a key role in guiding our interpretation and confirming the

spatial distribution of cultivated pastures. Furthermore, the accuracy of our classification is supported by a quantitative assessment, which yielded an overall accuracy of 97.05% ± 0.4% and an F1 spatial consistency score of 82.51% ± 14.22% (Precision: 90.04% ± 6.18%; Recall: 76.74% ± 9.91%).

b) Although the study spans 34 years (from 1988 to 2021), the number of valid remote sensing observations in any given year can exceed 34 due to the acquisition characteristics of the Landsat program. Landsat satellites have a nominal 16-day revisit cycle, and image swath overlaps can result in multiple acquisitions for a single location. Additionally, despite the presence of clouds and cloud shadows, a substantial number of high-quality observations may still be available in a single year. Therefore, the maximum number of valid observations shown in Figure 5/6b for 2021 may exceed 34 and reflects intra-annual data availability rather than the number of years in the study period.

Line273: a) In Table4, the total proportion of cultivated pastures and other types in Qinghai Province does not add up to 1. This inconsistency need to be clarified by the authors. b) The overall accuracy of 97.05% ± 0.4% mentioned in the abstract and discussion is only presented in those sections and not in the main text. Moreover, this accuracy represents the model's performance rather than the dataset's inherent quality. Moreover, they should also address the difference between model accuracy and dataset precision to ensure the reliability of the study's results.

Response: We sincerely thank the reviewer for the careful reading and thoughtful comments. We have addressed the two concerns as follows:

a) We appreciate the reviewer's attention to detail. The inconsistency in Table 4 was due to a typographical error—specifically, the value "0.072" should have been "0.134." We have corrected this error in the revised manuscript using the "Track Changes" feature in Word. We apologize for the oversight and thank the reviewer for bringing it to our attention.

b) The overall accuracy of 97.05% ± 0.4% was derived by aggregating all independent validation points from both Qinghai and Tibet. To improve clarity and transparency, we have added the following sentence to the main text of the manuscript:

> *"To evaluate the overall accuracy of the cultivated pasture mapping, we combined the validation points from Qinghai and Tibet, resulting in an overall accuracy of 97.05% ± 0.4% and an F1 spatial consistency score of 82.51% ± 14.22% (Precision: 90.04% ± 6.18%; Recall: 76.74% ± 9.91%)."*

We also appreciate the reviewer's observation regarding the distinction between model accuracy and dataset precision. In this study, our primary objective was to assess the quality of the final cultivated pasture dataset, rather than the performance of the classification model per se. Therefore, we employed an independent validation approach using a separate set of reference data, instead of cross-validation. This approach is better suited for evaluating the accuracy and reliability

of the dataset itself. We have added a clarifying statement in the revised manuscript to highlight this distinction and to avoid any potential confusion between model performance metrics and dataset validation.

Line294: a) Figure 10 shows that many areas in Tibet have cultivated pastures with a record time of less than 5 years, which seems inconsistent with general expectations. The authors should verify this phenomenon and provide a reasonable explanation. b) Figures 9 and S3 show that artificial grasslands suddenly disappear and reappear in the following year, which is inconsistent with the conventional pattern of grassland resource use. The authors should explain the reasons for this phenomenon.

Response: We appreciate the reviewer's comment. We have addressed the two concerns as follows.

a) Many areas in Tibet have cultivated pastures that have been established for less than five years. This observation aligns with our fieldwork experience in the region. Accordingly, we have added the following sentence to the manuscript: "Some of the cultivated pastures were established in very recent years, coinciding with the introduction of regional farming policies promoting the development of cultivated pastures (Fig. S4)."

b) This phenomenon aligns with the inter-annual crop rotation practices observed on cultivated pastures across the Tibetan Plateau, as evidenced by our field campaign experience in the region. On the Tibetan Plateau, cultivated pastures growing annual oats and barley typically alternate with conventional croplands every other year to nourish the soil. We also noted this in lines 337–338: *"In addition, we only used satellite remote sensing data from one year instead of multiple years (Potapov et al. 2022), because inter-annual crop rotation exists on cultivated pastures on the Tibetan Plateau."*

Line 295-298: a) The authors only mention that the study results are close to government statistics between 2010 and 2017, but do not provide a quantitative error analysis. It is recommended that the authors calculate error indicators such as MAE and MRE to more clearly evaluate data accuracy. b) In Table S1, the authors compare the results of their dataset with government statistics at the county level. The table shows significant differences between the two, such as an MAE of $3.24\times10^4$ and an MRE of 223% in some counties. Such high errors cast doubt on the data quality. The authors should further analyze and explain these discrepancies to enhance the credibility of their study.

Response: We thank the reviewer for the constructive comments, which prompted deeper reflection on our findings and contributed to strengthening our argument.

a) After carefully inspecting Fig. 11b, we observed a sharp increase in the reported area of cultivated pastures for Qinghai in the 2013 government statistics, suggesting a possible shift in the statistical criteria adopted that year. As government statistics are not necessarily definitive, we did not conduct a quantitative comparison between them and our results. We appreciate the reviewer's suggestion, which helped improve our argument. The following text has been added to the manuscript:

*"A sharp increase in the area of cultivated pastures for Qinghai is reported in the 2013 government statistics (Fig. 11b), suggesting a potential shift in the statistical criteria for cultivated pastures that year. In contrast, our results show a more gradual increase, which likely reflects a more accurate representation of the actual expansion of cultivated pastures on the Plateau. Our findings indicate that government statistics warrant further scrutiny in future policy development related to cultivated pastures."*

b)  Since the statistical criteria for cultivated pastures used by local governments do not fully align with our definition, we focused the comparison between our results and government statistics at the county level on correlation metrics rather than absolute or relative errors (Fig. 8 and Table S1). The coefficients of determination were 0.75 for Qinghai and 0.77 for Tibet, indicating the reliability of our results. We have revised our manuscript accordingly: *"Since the statistical criteria for cultivated pastures used by local governments do not fully align with our definition, we focused the comparison between our results and government statistics at the county level on correlation metrics rather than absolute or relative errors (Fig. 8 and Table S1). The coefficients of determination were 0.75 for Qinghai and 0.77 for Tibet, indicating the reliability of our results."*

Line363-367: Comparing data from 2021 with government statistics from 2017 is clearly unreasonable.

Response: We appreciate the reviewer's comment. The rationale for comparing data from different years is explained in lines 367–368: *"We used the statistics data of cultivated pastures in 2017 because it was the closest year in which the statical caliber used by the annual China Pratacultural Statistics agreed with our definition of cultivated pastures."* While we acknowledge that this comparison is not ideal, it represents the best available approach given the data constraints.

Line381-395: The discussion of the limitations of this study by the authors is not in-depth enough. Several key aspects require more thorough analysis and discussion, such as the technical limitations related to remote sensing data and modeling process.

Response: We sincerely thank the reviewer for the constructive and insightful feedback. In response, we have expanded the discussion of the limitations of our study to address the technical challenges related to both the remote sensing data and the modeling process. The revised Section 5.4 now includes a more detailed analysis of these limitations. Please see the updated text below.

**Revised Section 5.4 Limitations and future prospects**
The time series of cultivated pasture maps on the Tibetan Plateau were produced for the first time; nevertheless, the mapping method and the maps had several limitations and could be improved in the future.

(1) The remote sensing data used in this study were Landsat data with a spatial resolution of 30 m. While Landsat data have been widely utilized for land cover classification, the 30 m spatial resolution may be insufficient for

accurately capturing small cultivated pastures with dimensions smaller than 30 m. Specifically, long and narrow cultivated pastures, often found on the Tibetan Plateau, may not be well-represented in the 30-meter resolution map. In some cases, these small patches were adjacent to conventional croplands, leading to mixed pixel problems. When the proportion of cultivated pastures in these mixed pixels was less than 50%, the pixels were typically classified as croplands. This issue could potentially be alleviated with higher spatial resolution data, such as Sentinel-2 imagery, which has a spatial resolution of 10 m and became available in 2015 (Phiri et al., 2020). However, we opted not to use Sentinel-2 data, as our focus was on the long-term spatial distribution of cultivated pastures on the Tibetan Plateau.

(2) We employed quantile metrics of the remote sensing time series for cultivated pasture classification, a method that has proven successful in capturing cultivated pasture dynamics. The input features derived from these metrics were more effective compared to those used in previous studies, such as Wang et al. (2022), which relied on monthly NDVI, EVI, NDPI, SR, and SAVI data for select months. While the latter approach utilized limited spectral information, our method incorporated a broader range of spectral indices, thus enhancing the overall classification accuracy.

(3) As with many optical remote sensing studies, our research was affected by atmospheric disturbances such as cloud cover and cloud shadows, which can reduce the number of valid observations, particularly in certain regions and time periods. While we incorporated all available Landsat data during the growing season, the quality and density of the time series varied spatially and temporally, which may have impacted the consistency of the analysis.

(4) To manage the computational burden associated with processing extensive time series data, we utilized descriptive statistical metrics (i.e., 25%, 50%, and 75% quantiles) of the remote sensing data for the growing season. While this approach helped maintain classification accuracy while reducing computation time, it may have resulted in the loss of finer temporal phenological details that could be captured through the analysis of the full time series.

(5) The Random Forest (RF) algorithm, which we employed for binary classification, is a powerful method known for its ability to handle complex relationships in the data. However, its performance heavily depends on the quality and representativeness of the training data. We are confident that the large volume of training data collected during extensive fieldwork, which encompassed a variety of cultivated pasture types, contributed significantly to model performance.

(6) Although we validated the cultivated pasture maps for 2021 in two pilot study regions with different climates, landscapes, soil properties, and ecological conditions, and observed a good match with government statistics at both the county and provincial levels, the validation for maps spanning 1988 to 2021 could benefit from further feedback. Additional validation efforts by

other researchers or practitioners in different regions and under varying local conditions could provide important insights for refining and improving the cultivated pasture mapping methodology.

Minor concerns:

Line25: What does F1 refer to?

Response: The F1 measure, defined as the harmonic mean of precision and recall, is a fundamental metric for assessing the accuracy and balance of binary classification models. It has been widely used in previous land cover classification studies, including two recent land cover classification studies in Earth System Science Data (ESSD).

Mei, Qinghang, Zhao Zhang, Jichong Han, Jie Song, Jinwei Dong, Huaqing Wu, Jialu Xu, and Fulu Tao. 2024. "ChinaSoyArea10m: A Dataset of Soybean-Planting Areas with a Spatial Resolution of 10 m across China from 2017 to 2021." Earth System Science Data 16, no. 7: 3213–31. https://doi.org/10.5194/essd-16-3213-2024.

Tommaso, Stefania Di, Sherrie Wang, Rob Strey, and David B. Lobell. 2024. "Mapping Sugarcane Globally at 10 m Resolution Using Global Ecosystem Dynamics Investigation (GEDI) and Sentinel-2." Earth System Science Data 16, no. 10: 4931–47. https://doi.org/10.5194/essd-16-4931-2024.

Line326: The authors' discussion of the methods used to map cultivated pastures is too superficial.

Response: We appreciate the reviewer's insightful comment. In response, we have substantially revised Section 5.1 of the Discussion to provide a more in-depth explanation of the methodological innovations and rationale behind our mapping approach. Specifically, we now elaborate on the advantages of using spectral-temporal metrics over complete time series data, the implications of training data derived from polygons versus points, the justification for using single-year imagery in the context of crop rotation, and the methodological rigor of our independent validation strategy. These additions are intended to clarify the methodological contributions of our work and to provide a more critical comparison with existing approaches in the literature. Please see the revised Section 5.1 below.

**Revised Section 5.1 The method for cultivated pasture mapping**

A distinctive feature of our mapping method is the use of spectral-temporal metrics derived from remote sensing time series data, rather than the complete time series of all valid observations during the growing season. While prior studies (e.g., Wang et al., 2022) have relied on dense time series data to characterize vegetation dynamics, our approach condenses these data into a set of statistical descriptors (e.g., median, maximum, minimum, standard deviation) of key vegetation indices. These spectral-temporal metrics serve as compact representations of phenological patterns and temporal variability in vegetation reflectance, which are especially informative in distinguishing cultivated pastures from other land cover types. This strategy reduces data dimensionality and computational load while retaining the essential temporal information relevant for classification, thereby enhancing both efficiency and accuracy.

Another critical strength of our approach lies in the volume and structure of the training data. Our dataset was built from training polygons collected through extensive fieldwork, rather than from isolated training points. Polygons offer a more comprehensive sampling of spectral variability within each land cover type and provide more training samples to the classifier, improving generalization. In contrast, previous studies (e.g., Wang et al., 2022) often used sparse point-based training data, which may not adequately capture the heterogeneity of cultivated pasture across large regions.

Furthermore, we chose to use remote sensing data from a single growing season rather than multi-year composite datasets (e.g., Potapov et al., 2022). This decision was informed by the unique land management practices on the Tibetan Plateau, where inter-annual crop rotation is prevalent among cultivated pastures. Although multi-year composites are useful in mitigating atmospheric noise such as cloud contamination, they risk introducing classification errors due to the temporal inconsistency of land cover resulting from rotation. By focusing on a single year, we ensure that the remote sensing signatures align with the actual land cover state at the time of classification.

Finally, we implemented an independent validation strategy based on an external reference dataset (Yang et al., 2021) rather than relying on internal cross-validation. While cross-validation is common in remote sensing applications (Ashourloo et al., 2018; Wang et al., 2022), it can lead to over-optimistic accuracy estimates if the same spatial samples are used for both training and validation (Friedl et al., 2000; Foody, 2002). By separating the training and validation datasets, our assessment provides a more realistic and conservative estimate of classification performance, especially across diverse pasture types.

Line341-344: What are the ratios of independent validation, training samples, and validation samples?
Response: We appreciate the reviewer's comment. In this study, we used independent randomly selected validation samples that were entirely separate from the training samples. Therefore, the ratio between training and validation samples did not affect our analysis.

Figure S1: Why do the number of good observations show such distinct striping patterns?
Response: We appreciate the reviewer's comment. The striping patterns arise from the overlapping paths of Landsat swaths and have been documented in key studies by members of the Landsat Science Team. We added the following sentence to the manuscripts: *"The striping patterns in Figure S1 arise from the overlapping paths of Landsat swaths (Zhang et al. 2022)."*

Zhang, Yingtong, Curtis E. Woodcock, Paulo Arévalo, Pontus Olofsson, Xiaojing Tang, Radost
Stanimirova, Eric Bullock, Katelyn R. Tarrio, Zhe Zhu, and Mark A. Friedl. 2022. "A Global Analysis
of the Spatial and Temporal Variability of Usable Landsat Observations at the Pixel Scale." Frontiers
in Remote Sensing 3: 894618. https://doi.org/10.3389/frsen.2022.894618.